# Antifungal Drugs

**DOI:** 10.3390/metabo10030106

**Published:** 2020-03-12

**Authors:** Jiří Houšť, Jaroslav Spížek, Vladimír Havlíček

**Affiliations:** Institute of Microbiology of the Czech Academy of Sciences, Vídeňská 1083, 14220 Prague, Czech Republic; jiri.houst@biomed.cas.cz (J.H.); spizek@biomed.cas.cz (J.S.)

**Keywords:** antifungal drugs, amphotericin B, flucytosine, triazoles, echinocandins, invasive fungal infections, resistance, siderophores

## Abstract

We reviewed the licensed antifungal drugs and summarized their mechanisms of action, pharmacological profiles, and susceptibility to specific fungi. Approved antimycotics inhibit 1,3-β-d-glucan synthase, lanosterol 14-α-demethylase, protein, and deoxyribonucleic acid biosynthesis, or sequestrate ergosterol. Their most severe side effects are hepatotoxicity, nephrotoxicity, and myelotoxicity. Whereas triazoles exhibit the most significant drug–drug interactions, echinocandins exhibit almost none. The antifungal resistance may be developed across most pathogens and includes drug target overexpression, efflux pump activation, and amino acid substitution. The experimental antifungal drugs in clinical trials are also reviewed. Siderophores in the Trojan horse approach or the application of siderophore biosynthesis enzyme inhibitors represent the most promising emerging antifungal therapies.

## 1. Introduction

Invasive fungal infections represent a global problem resulting in 1.7 million deaths every year [1,2]. They are common in immunocompromised patients, as reflected in their chemotherapy, acquired immune deficiency syndrome, and/or organ transplantation [1]. The recent annual incidence of invasive aspergillosis, candidiasis, and mucormycosis is over 300,000, 750,000, and 10,000 cases, respectively [3]. The incidence of mucormycosis may exceed 900,000 cases per year after the inclusion of Indian data estimates [4]. Furthermore, these infections are associated with high mortality rates. The epidemiology of invasive fungal infections usually focuses on specific areas. The lack of available global data leads to a broad range of mortality rates, e.g., 30%–95% and 46%–75% in invasive aspergillosis and candidiasis, respectively [5]. The overall incidence of disseminated scedosporiosis and fusariosis is one [6] or six [7] cases per 1000 hematopoietic stem cell transplant recipients.

Currently, four antifungal drug classes are used by clinicians and veterinarians for systemic treatment [8]. These classes target different parts of the fungal cell. First, the polyene class includes the heptaene amphotericin B (AMB), which interacts with ergosterol, the major part of the fungal cell membrane. AMB is highly fungicidal against *Candida* genera [9] and *Aspergillus fumigatus* and *A. flavus* [10]. Second, first- and second-generation of triazoles disrupt the ergosterol biosynthesis in the lanosterol demethylation step. Generally, triazoles exhibit the fungistatic effect against yeasts but are fungicidal for *Aspergillus* spp. [11]. Echinocandins block the synthesis of β-d-glucans located in the fungal cell wall. Echinocandins are fungicidal and fungistatic against *Candida* and *Aspergillus* spp., respectively [12]. Finally, the pyrimidine analogue flucytosine (5-FC) interacts at the nucleus level of the fungus, affecting protein and deoxyribonucleic acid (DNA) biosynthesis [8]. The overuse of antifungal agents increases the opportunistic pathogen resistance [13]. The World Health Organization has identified this type of antimicrobial resistance as one of the dominant threats of 2019 [14].

In this work, we reviewed the important approved and selected experimental antifungal drugs. Immunomodulatory therapies [15,16], covering both molecular and cell-based therapies, were not the subject of this manuscript. Similarly, the application of mycoviruses or therapeutic enzymes to degrade fungal biofilms or cell wall structures has not been included in the present communication and can be found elsewhere [17,18].

## 2. Overview of Antifungal Agents and Their Mechanisms of Action

Since the 1950s, more than 200 polyenes with antifungal activity have been discovered; however, amphotericin B remains the single polyene drug of choice in the treatment of invasive fungal infections [19]. 5-Flucytosine was first designed and used as the antimetabolite in cancer treatment. As its antineoplastic activity was low, 5-FC now serves in combinational antifungal therapy [20]. Whereas fluconazole (FLC) and itraconazole (ITC) are first-generation triazoles, the second-generation of triazoles, with improved pharmacological profiles, include voriconazole (VOR), posaconazole (POS), and isavuconazole (ISV) [21]. Caspofungin (CSF), micafungin (MCF), and anidulafungin (ANF) represent the newest class of peptide antifungals called echinocandins. The chronology of antifungal application and development is shown in Figure 1. The power of combination therapy in the treatment of invasive fungal infections was summarized recently [22].

Amphotericin B is a cyclic heptaene (Figure 2A) produced by the Gram-positive bacterium *Streptomyces nodosus*. It has two mechanisms of action (Figure 3). First, several molecules of AMB incorporate into the fungal lipid bilayer and bind to ergosterol. By ergosterol sequestration, pores are formed, and both the ions (K^+^, Mg^2+^, Ca^2+^, and Cl^−^) and electrolyte glucose are released. The rapid depletion of intracellular ions results in fungal cell death. Second, AMB induces the accumulation of reactive oxygen species (ROS), resulting in DNA, protein, mitochondrial, and membrane damage [23]. The structural parts mycosamine and hydroxyl groups at the C8, C9, and C35 positions are critical for the AMB biological activity [24].

5-Flucytosine (Figure 2B) is a synthetic analogue of cytosine. After administration, 5-FC is taken up by cytosine permease into the fungal cell and deaminated to 5-fluorouracil (Figure 2C) by cytosine deaminase. 5-Fluorouracil is subsequently converted into 5-fluorouridine triphosphate. Unlike uridylic acid, this compound is incorporated into fungal ribonucleic acid (RNA), resulting in inhibition of protein synthesis (Figure 3). Besides, 5-fluorouracil can be metabolized to the 5-fluorodeoxyuridine monophosphate by uridine monophosphate pyrophosphorylase. This compound inhibits the primary source of thymidine in DNA biosynthesis, thymidylate synthetase, due to the enzyme’s inability to remove the fluorine atom [25].

The mechanism of action of triazoles (Figure 2D–H) is based on the inhibition of the microsomal cytochrome P450 (CYP450) monooxygenase dependent 14-α-demethylase (Figure 3). The demethylation of fungal lanosterol is a two-step process involving the reduced form of nicotinamide dinucleotide phosphate (NADPH) and oxygen. As nitrogen from the triazole ring binds to the heme iron, oxidation of the methyl group is prevented. The combination of the accumulation of toxic 14-α-methylsterols and depletion of ergosterol results in the fungistatic effect [26].

Echinocandins are modified analogues of pneumocandins and are fermentation products of a variety of microorganisms. Namely, anidulafungin is derived from echinocandin B_0_ (*Aspergillus nidulans*), caspofungin from pneumocandin B_0_ (*Glarea lozoyensis*), and micafungin from pneumocandin A_0_ (*Coleophoma empetri*). Echinocandins are targeted against 1,3-β-d-glucan synthase (Figure 3), the enzyme complex composed of the transmembrane catalytic Fks and intracellular regulatory Rho1 subunits. The former is non-competitively inhibited; therefore, 1,3-β-d-glucan synthase cannot convert uridine diphosphate glucose to a β-d-glucan, and the fungal cell wall becomes highly permeable [27]. Echinocandins are cyclic hexapeptides with specific lipophilic *N*-acetylated side chains (Figure 4A–C). Their core determines their physicochemical properties—homotyrosine inhibits 1,3-β-d-glucan synthase; proline residues enhance antifungal potency; and substitution with a hydroxyl group, ethylenediamine, and sulphated moieties improve water solubility. Because the linoleoyl side chain in echinocandin B_0_ exhibit the hemolytic activity, it has been modified to a dimethylmyristoyl, diphenyl substituted isoxazole, and alkoxytriphenyl chain [12,28].

## 3. Pharmacology and Toxicity of Antifungal Agents

In the drug formulation, amphotericin B is bound to sodium deoxycholate because of its limited solubility in water. After intravenous administration, AMB dissociates from deoxycholate, binds to plasma lipoproteins, and accumulates in the spleen and liver. With the elimination half-life of over 15 days, AMB is not metabolized by CYP450 enzymes, but is excreted into the urine (33%) and feces (43%) as the unchanged drug. The inherent limitations of AMB deoxycholate application consist in its infusion- and dose-related toxicity. The infusion-related toxicity arises from the induction of pro-inflammatory cytokines and chemokines such as interleukin 1β, tumor necrosis factor α, monocyte chemotactic protein 1, and macrophage inflammatory protein 1β. On the other hand, high doses of AMB cause nephrotoxicity due to the non-selective disruption of renal cells. To reduce these undesirable effects, AMB deoxycholate has been incorporated into lipidic formulations still retaining its fungicidal activity. Three lipidic formulations are commercially available: AMB lipid complex (ABLC), liposomal AMB (LAMB), and AMB colloidal dispersion (ABCD) [29].

In contrast, 5-flucytosine, as a small hydrophilic molecule, is absorbed rapidly, and its bioavailability reaches almost 90%. Liver enzymes metabolize 5-FC minimally. It is eliminated exclusively via glomerular filtration with excellent antifungal activity in the bladder, and its plasma clearance is as fast as creatinine clearance. 5-FC is dosed frequently because of its half-life, which is up to four hours. Co-administration of 5-FC with the cytarabine (an effective drug for the treatment of acute myeloid leukemia [30]) competitively inhibits its antifungal activity due to the same transport system by susceptible cells. Severe side effects include hepatotoxicity, myelotoxicity, and gastrointestinal problems [20].

The pharmacological profiles of triazoles vary with their molecular weight, solubility, and protein binding. Selected pharmacological data are presented in Table 1. All triazoles are available both for peroral and intravenous applications. Notably, isavuconazole is administered as the water-soluble prodrug isavuconazonium. Furthermore, the only metabolism of itraconazole results in the active metabolite hydroxyitraconazole.

On the other hand, genetic polymorphisms of CYP2C19 and CYP3A4 affect the metabolism of voriconazole. Generally, triazoles are well tolerated. The most common serious side effect, hepatotoxicity, occurs most often with VOR (in 31% cases). Whereas ISV shortens the QT interval (the time between the start of the ventricular depolarization and the end of the ventricular repolarization), the rest of triazoles act in the opposite way. Unfortunately, triazoles have many potential drug–drug interactions due to their affinity for CYP450 isoenzymes [31].

Echinocandins have so far been approved for intravenous administration only. Selected pharmacological data are presented in Table 2. High protein binding and negligible metabolism by CYP450 are common among them; on the other hand, their half-life and degradation processes are different. Caspofungin has the lowest half-life, and after the spontaneous opening of the peptide ring, it is further degraded by both peptide hydrolysis and *N*-acetylation into two inactive metabolites. In contrast, anidulafungin has the longest elimination half-life. Its biotransformation results in an ineffective open-ring peptide. Finally, the metabolism of micafungin forms three metabolites by arylsulphatase, catechol-*O*-methyltransferase (COMT), and CYP3A side-chain hydroxylation (Figure 5). Only the degradation products of CSF are excreted primarily into the urine. Echinocandins have few known drug–drug interactions because of their reduced substrate potential to CYP450 enzymes [37].

Adequate antifungal dosing strategies are necessary for ensuring successful treatment. The optimal antifungal dosages vary by given patient populations and include both patient physical condition and type of treatment. Therefore, dosage recommendations are written in several guidelines and review articles, including aspergillosis and candidiasis in neonates [42], pediatrics [43], and adults [44,45]. These recommendations are summarized and simplified in Table 3. Furthermore, antifungal prescription for pregnant women is a challenge due to the limited information about embryotoxic/teratogenic effects. According to the Food and Drug Administration classification, only AMB is considered as the safest drug for the treatment of systemic fungal infection. The rest of antifungals has a worse reputation because of indication and/or positive evidence of fetal risk based on animal studies [46]. Dose improvement is also necessary for the obese population due to their often exclusion from drug development studies resulting in limited pharmacological data for antifungals. Whereas FLC, CSF, and MCF are correlated with total body weight, dosing alterations are not necessary for POS, AMB, and ANF [47].

## 4. Susceptibility of Antifungal Agents

Antifungal susceptibility testing (AST) is a must for both the optimal treatment and the detection of possible antifungal resistance. AST is performed by both reference and commercial methods based on broth microdilution and agar-based (or disk diffusion) assays. These assays are standardized according to the European Committee on Antimicrobial Susceptibility Testing (EUCAST) [49] and the Clinical and Laboratory Standard Institute (CLSI) [50]. Although the reference broth microdilution methods are the gold standard for AST, they are time-consuming and laborious. Therefore, commercial methods have been developed for daily AST. These methods are automated or semi-automated, rapid, and low cost [51].

To distinguish if the microorganism isolates are resistant to antifungal agents or not, EUCAST and CLSI have set clinical breakpoints expressed as the minimal inhibitory concentration (MIC), indicating the minimum drug concentration that inhibits fungal growth. According to the 2019 EUCAST definition [52], a microorganism can be categorized into the S (susceptible, standard dose regime), I (susceptible, increased exposure), or R (resistant) category. Additionally, CLSI distinguishes between SDD (susceptible, dose-dependent) and NS (non-susceptible) category [53]. Despite the slight methodological differences, both CLSI and EUCAST yield comparable MIC data. Unluckily, CLSI has not determined the clinical breakpoints against any molds [54]. Therefore, modified data presented in Table 4 and Table 5 summarize the susceptibilities of antifungal agents against selected *Aspergillus* and *Candida* species covering the period 2007–2019 from EUCAST only [55].

The European Society for Clinical Microbiology and Infectious Diseases (ESCMID) published guidelines for the management of *Aspergillus* [45], *Candida* [44], *Fusarium,* and *Scedosporium* [56] infections. In these guidelines, the correct selection of an antifungal agent is based on the clinical experience, case studies, clinical trials, and population of patients. Therefore, voriconazole is also recommended for the management of aspergillosis caused by *A. terreus* and *A. niger*. Furthermore, VOR should be used in both first-line and salvage therapy in immunocompromised patients suffering from fusariosis and scedosporiosis. In the case of candidemia and invasive candidiasis, the use of all three echinocandins (including caspofungin) is strongly recommended as an initial targeted treatment in the adult population.

## 5. Resistance to Antifungal Agents

Microbial resistance to antifungal drugs is a result of multiple factors and emerges by a series of molecular mechanisms (Figure 6). Whereas some intrinsic resistance has been found naturally, e.g., fluconazole-resistant *C. krusei* [57], *C. glabrata* [58], and *Aspergillus* species [59], the acquired resistance has been a consequence of long-term therapies, widespread prophylaxis, or use of antifungals in agriculture, especially in the case of triazoles [60]. Environmental exposure of *A. fumigatus* to triazole fungicides may explain their resistance in azole-naïve patients [61]. Additionally, secondary resistance may occur after vertical and horizontal transmission in both animals [62] and humans [63].

Amphotericin B susceptibility depends on the ergosterol content in the fungal cell membrane. Ergosterol biosynthesis is regulated by 25 known *ERG* enzymes [66]; their alterations (*ERG3*, *ERG5*, *ERG11*), gene deletion (*ERG11*), and harbored mutations (*ERG1*, *ERG2*, *ERG6*, *ERG11*) lead to the decreased AMB sensitivity in *Candida* species [67]. Furthermore, resistant *C. tropicalis* isolates showed a thickened cell wall due to the increased content of 1,3-β-d-glucans [68]. On the other hand, the higher activity of catalase and superoxide dismutase and the more intense stress response through heat shock proteins 70 and 90 (Hsp70, Hsp90) contribute to the intrinsic resistance of *A. terreus* [69].

5-Flucytosine is known for its rapid development of resistance; therefore, it is used only in combination therapy with AMB and triazoles. Primary and secondary resistance in clinically relevant *Candida* spp. have emerged as a consequence of alterations in the *FCY2*, *FCY1*, and *FUR1* genes responsible for 5-FC uptake and its conversion, respectively. Additionally, the susceptibility of *C. glabrata* to 5-FC increased after deletion of the *CgFPS1* and *CgFPS2* homologue encoding plasma membrane aquaglyceroporins that are required for osmotic stress resistance [70]. It has been shown that 5-FC activity increases up to 4000-fold against *A. fumigatus* at pH 5. Furthermore, *FCYB* gene expression is repressed by the cytosine-cytosine-adenosine-adenosine-thymidine binding complex (CBC) and pH-response transcriptional factor PacC at pH 7, confirming intrinsic resistance [71].

The first mechanism of azole resistance involves amino acid substitutions near the heme binding site of 14-α-demethylase or overexpression of the *ERG11*/*CYP51A*/*CYP51B* genes included in ergosterol biosynthesis. The overexpression of *ERG11* is due to gain-of-function mutations in Upc2 in *C. albicans*. Furthermore, another transcriptional factor SrbA is required for azole resistance in *A. fumigatus*. The second mechanism of acquired azole resistance arises from the upregulation of ABC (adenosine triphosphate binding cassette) transporters as well as major facilitators responsible for higher drug efflux [13]. Cdr1, Cdr2, and Mdr1 are responsible for azole efflux in *C. albicans*, and AtrF, Mdr3, and Mdr4 in *A. fumigatus*, respectively. Furthermore, intrinsic resistance to triazoles is often caused by *Candida* biofilm formation [72].

The basis of echinocandin resistance consists of amino acid substitutions in the *FKS1* and *FKS2* subunits of glucan synthase. The most frequent amino acid changes encompass Ser641 and Ser645 in *C. albicans* and Ser629, Phe659, and Ser663 in *C. glabrata*. Moreover, the elevated MIC values in *C. parapsilosis* are due to naturally occurring polymorphisms linked with the proline-to-alanine change in *FKS1*. *FKS2*-dependent resistance in *C. glabrata* can be reversed by treatment with the calcineurin inhibitor tacrolimus. Treatment with echinocandins is connected with higher biosynthesis of β-d-glucans and chitin in *C. glabrata* and *C. parapsilosis*. The higher molecular weight cell wall content may play a role in echinocandin resistance [73]. In addition, protection against cell wall weakening is induced through a variety of stress adaption mechanisms involving protein kinase C, calcineurin, and Hsp90 [74].

The clinically relevant *Fusarium* spp. (especially *F. solani* complex, *F. oxysporum*, and *F. fujikuroi* complex [75]) are resistant to amphotericin B, triazoles, and echinocandins. In the case of triazoles, the combination of *CYP51A* amino acid substitution and/or gene overexpression may be involved. In *FKS1*, the changes of Phe639 to Tyr and Pro647 to Ala contribute to echinocandin resistance. Moreover, cross-resistance has been observed among both triazoles and echinocandins [76]. The *Scedosporium* species have intrinsically low susceptibility or high resistance to azoles; furthermore, their echinocandin resistance is due to the Trp-to-Phe substitution at position 695 [77]. *Mucormycetes* are resistant to short-tailed triazoles voriconazole and fluconazole. The molecular mechanism resistance comprises the Phe129 substitution for Tyr in the paralogue gene *CYP51 F5*, including the possible contribution of Val-to-Ala substitutions at positions 291 and 293. On the other hand, isavuconazole, posaconazole, and itraconazole usually provide effective treatment [78].

## 6. The Antifungal Pipeline

The research in antifungal drug development focuses on both old and new fungal targets. In addition, antifungal susceptibility testing and ongoing clinical trials have modified already licensed antifungals, i.e., drugs with originally different purposes or novel synthesized substances (Table 6).

In ergosterol biosynthesis, modified tetrazoles (VT-1129, 1161, 1598; Figure 7A–C) show higher specificity to the fungal Cyp51 [84]. The new triazole PC1244 (Figure 7D) has recently been reported and is specifically designed to be inhaled and persists at high concentrations in the lungs for a long time [82]. To enhance bioavailability of itraconazole after oral administration, new super bioavailable itraconazole (SUBA-ITC) has been developed. This SUBA-ITC comprises its solid dispersion in a pH-dependent matrix ensuring doubled bioavailability compared to the original ITC formulation [85]. Furthermore, coumarins and aminopiperidines as non-azole compounds inhibit *ERG11* and *ERG24*, respectively. Aminopiperidines show antifungal activity against fluconazole-resistant *Candida* spp. and *A. fumigatus* [86]. In addition to ergosterol sequestration, amphotericin B cochleate (CAMB) has been developed as an oral formulation of AMB. CAMB is made up of phosphatidylserine and phospholipid-calcium precipitates and demonstrates successful treatment in murine mouse model against *C. albicans* [85].

In β-d-glucan synthesis, newly developed molecules include rezafungin (CD101, Figure 7E) with an extended half-life (>80 h), enabling once-weekly intravenous dosing. Rezafungin is a peptide that has not yet passed all the phases of testing [22]. Ibrexafungerp (SCY-078, Figure 7F) is a triterpenoid and is one of the first glucan synthase inhibitor administered *per os* [84]. Both agents have low rates of drug–drug interactions and excellent safety profiles [23]. However, neither is active against mucormycetes.

Contemporary research is directed towards novel fungal targets, especially enzymes involved in other metabolic pathways [87]. The inositol acyltransferase inhibitor Fosmanogepix (APX001, Figure 7G) prevents microorganism colonization/adhesion [86] and is effective against pulmonary scedosporiosis or fusariosis in immunocompromised mice [88]. The dihydroorotate dehydrogenase inhibitor F901318 (Olorofim, Figure 7H), the first representative of the orotomides, inhibits the pyrimidine biosynthesis and is active against *Aspergillus* and *Scedosporium* spp., including *Lomentospora prolificans* [89]. Co-administration of Hos2 fungal histone deacetylase inhibitor MGCD290 with both azoles and echinocandins enhance fungicidal effect against *Candida* and *Aspergillus* spp. [85]. Arylamidine T-2307 (Figure 7I) is taken up via the spermine/spermidine transport system and probably affects mitochondrial membrane potential balance in yeast [83]. Furthermore, monoterpenoids citronellal (Figure 7J) and perillaldehyde (Figure 7K) enhance the ROS production in *C. albicans,* resulting in DNA and mitochondrial damage. In this pathogen, inhibition of the prostaglandin E_2_ by cyclooxygenase inhibitors reduces the biofilm development [90].

Acylhydrazones, inhibitors of fungal sphingolipid synthesis, showed promising results in terms of efficacy and toxicity. For example, brominated acylhydrazone BHBM (Figure 7L) and its derivative D13 (Figure 7M) were found to be highly effective against several pathogenic fungi, including *Cryptococcus neoformans, Cryptococcus gattii, Rhizopus oryzae, Histoplasma capsulatum, Blastomyces dermatitidis, Pneumocystis murina*, and *Pneumocystis jirovecii* [91,92]. More antifungal compounds in development with novel targets can be found in an excellent review by J.R. Perfect [87].

In particular, the basic experimental way of antimicrobial research focuses on siderophores. Iron hijack belongs to profound virulence factors in infection; therefore, disruption of iron assimilation offers three novel antimicrobial strategies. The first strategy uses the siderophore-antifungal drug-conjugate in a Trojan horse therapy that enables drug delivery into the cytoplasm of the microorganism and drug release at the active site. These conjugates arise naturally (sideromycins) or synthetically with antibiotics [93]. Next, inhibition of nonribosomal peptide synthetases (NRPS), polyketide synthases, and NRPS-independent siderophore synthetases prevents iron scavenging [94]. As an example, celastrol (Figure 7N) blocks *A. fumigatus* growth by the inhibition of flavin-dependent monooxygenase siderophore A required for the L-ornithine hydroxylation [95]. Last, competitive iron chelators, such as lactoferrin and iron substitution by gallium, can reduce biofilm formation [93]. Phosphopantetheine transferase inhibitors directly affecting siderophore synthesis also represent the possible future antifungals [96,97], either alone or in combination therapy.

It is worth mentioning that specialized mammalian cells secrete lipocalins preventing iron hijack by catecholate-type siderophore sequestration [98]. VL-2397 (Figure 7O) is a hydroxamate siderophore similar to ferrichrome. It was isolated from *Acremonium* spp. and exhibits significant activity against *Aspergillus* spp. The mechanism of action of this compound is not known. Still, its selectivity for fungal cells derives from the fact that it is taken up by the fungal siderophore transporter Sit1, which is absent from mammalian cells [99]. Besides these approaches, siderophores can lead to an early, selective, and sensitive diagnosis of infectious diseases [100], and their detection in urine is non-invasive [101].

## Figures and Tables

**Figure 1 metabolites-10-00106-f001:**
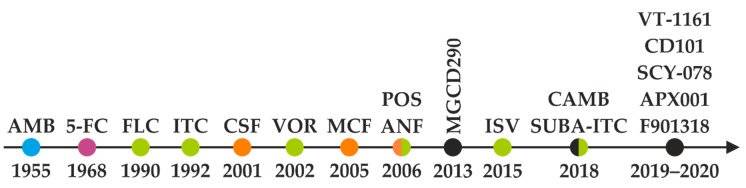
Sixty-five years of antifungal therapy. Most antifungal agents were discovered in the most recent three decades. Colored dots refer to specific antifungal drug class—polyenes (blue), pyrimidine analogues (purple), triazoles (green), and echinocandins (orange). Antifungal drugs under development and/or testing in clinical trials are marked black (see the chapter *Antifungal pipeline* for further information). AMB = amphotericin B, 5-FC = flucytosine, FLC = fluconazole, ITC = itraconazole, CSF = caspofungin, VOR = voriconazole, MCF = micafungin, POS = posaconazole, ANF = anidulafungin, ISV = isavuconazole, SUBA-ITC = super bioavailable itraconazole.

**Figure 2 metabolites-10-00106-f002:**
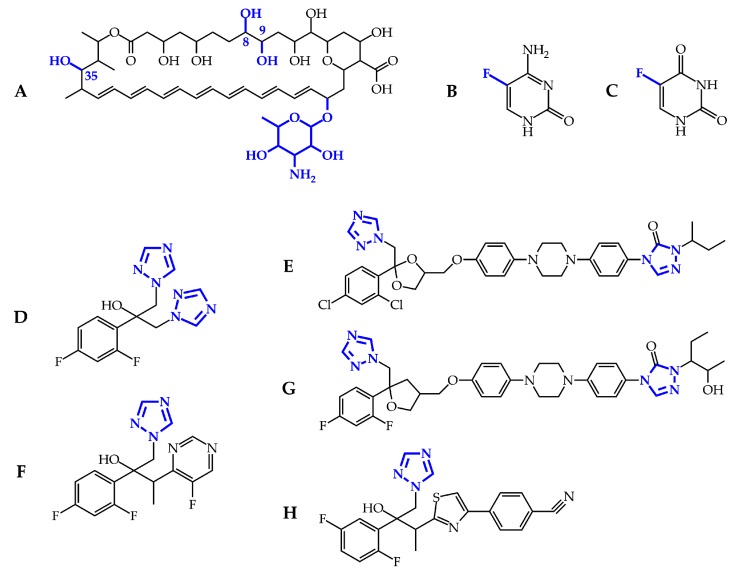
Chemical structures of the licensed antifungal drugs: amphotericin B (**A**); flucytosine (**B**), and its deaminated product 5-fluorouracil (**C**); fluconazole (**D**); itraconazole (**E**); voriconazole (**F**); posaconazole (**G**); and isavuconazole (**H**). The substructures necessary for biological activity are in blue.

**Figure 3 metabolites-10-00106-f003:**
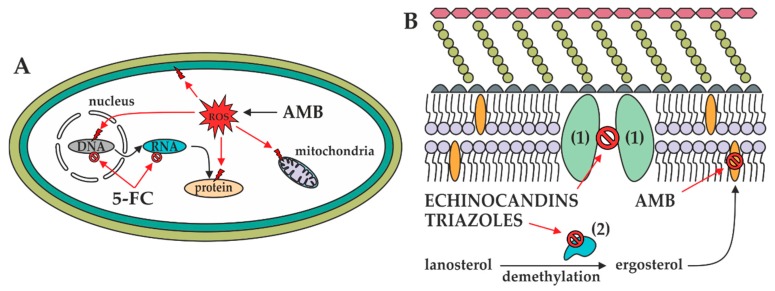
Mechanisms of actions of antifungals in the fungal cell. (**A**): In the fungal cytoplasm, AMB induces ROS formation resulting in the mitochondrial, biomembrane, DNA, and protein damage. Furthermore, 5-FC prevents DNA, RNA, and thus protein biosynthesis. (**B**): In the fungal biomembrane, 1,3-β-d-glucan synthase (1) and 14-α-demethylase (2) are inhibited by echinocandins and triazoles, respectively. Additionally, ergosterol contained in the biomembrane is sequestered by AMB, resulting in pore formation.

**Figure 4 metabolites-10-00106-f004:**
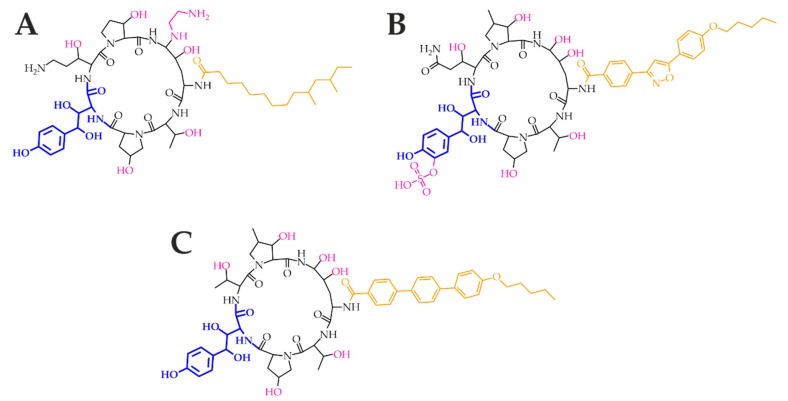
Echinocandins caspofungin (**A**), micafungin (**B**), and anidulafungin (**C**). The homotyrosine amino acid residue (blue) is mandatory for the inhibition of the 1,3-β-d-glucan synthase catalytic subunit Fks. The amino acid core (black) contributes to the antifungal potency and determines the physicochemical properties. Moreover, ethylenediamine, sulphate and hydroxyl groups (pink) contribute to water solubility. Specific lipophilic side chains (orange) decrease the hemolytic activity.

**Figure 5 metabolites-10-00106-f005:**
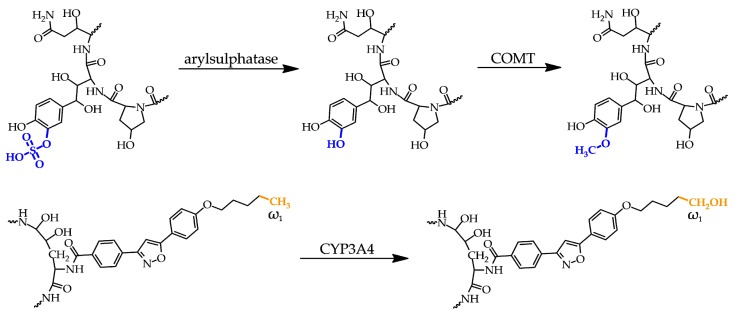
Metabolites of micafungin. Top: The original sulphate group (blue) on the dihydroxyhomotyrosine amino acid residue is metabolized to a hydroxyl group followed by its methylation. Bottom: the third and minor metabolite arises from hydroxylation of the methyl group (orange) at the ω_1_ position of the lipophilic side chain. COMT = catechol-*O*-methyltransferase.

**Figure 6 metabolites-10-00106-f006:**
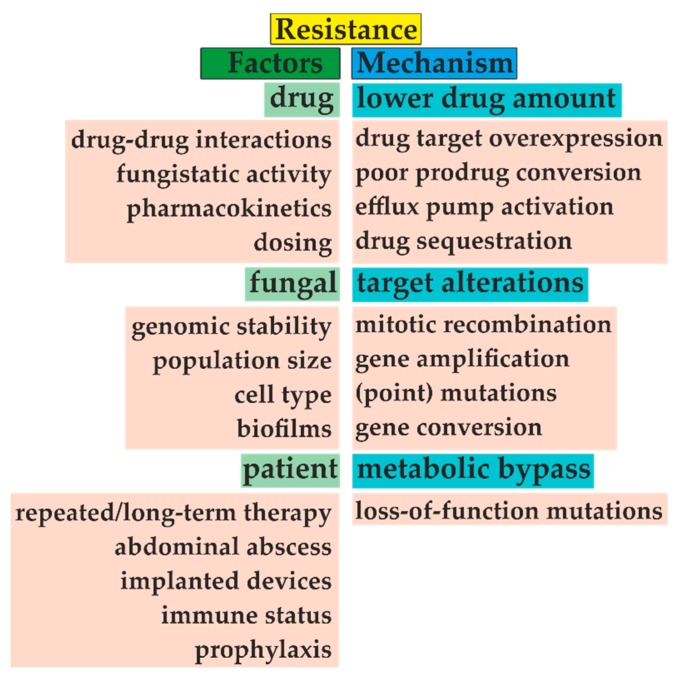
The emergence of microbial resistance is given by three main factors, including choice of antifungal treatment, type of fungal species, and patient medical history. For example, adequate dosing and distinguishing between fungistatic/fungicidal drug effects are mandatory for successful treatment. Unfortunately, repeated antifungal therapy and often prophylaxis narrow the appropriate drug selection. With the fungal biofilm formation, this task becomes more problematic. Furthermore, fungi often decrease drug concentration by efflux pump activation or target overexpression. Additionally, these targets can be amplified or changed due to several types of mutations, such as amino acid substitution. Modified from [64,65].

**Figure 7 metabolites-10-00106-f007:**
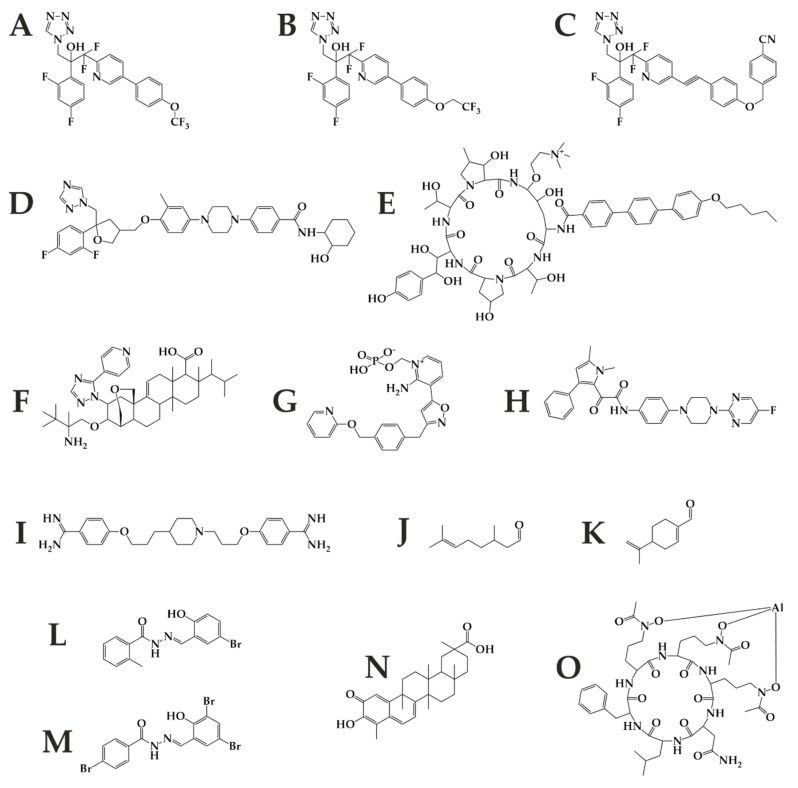
Future antifungal therapy may benefit from these substances including tetrazoles VT-1129 (**A**), VT-1161 (**B**), VT-1598 (**C**); triazole PC1244 (**D**); echinocandin rezafungin (**E**); triterpenoid Ibrexafungerp (**F**); inositol transferase inhibitor Fosmanogepix (**G**); dihydroorotate dehydrogenase inhibitor Olorofim (**H**); histone deacetylase inhibitor MGCD290 (**I**); monoterpenoids citronellal (**J**) and perillaldehyde (**K**); brominated acylhydrazones BHBM (**L**) and its derivative D13 (**M**); triterpenoid celastrol (**N**); and hydroxamate type siderophore VL-2397 (**O**).

**Table 1 metabolites-10-00106-t001:** Pharmacological properties of triazoles.

Triazole	FLC [32]	ITC [33]	VOR [34]	POS [35]	ISV [36]
Bioavailability [%]	90	55	90–96	variable	>98
Protein binding [%]	11–12	99	51–67	>98	>99
Metabolism (CYP)	–	3A4	2C19/2C9/3A4	glucuronidation	3A4/3A5
Half-life [h]	27–37	15–42	6	15–35	80–130
Route of elimination	renal	hepatic	hepatic	hepatic	hepatic
Unchanged [%]	64–90	3–18	<2	66	45

**Table 2 metabolites-10-00106-t002:** Pharmacological properties of echinocandins.

Echinocandin	ANF [38,39]	CSF [38,40]	MCF [38,41]
Bioavailability [%]	2–7 (p.o.), 100 (iv.)	100 (iv.)	100 (iv.)
Protein binding [%]	99	95	>99
Metabolism (CYP)	–	–	3A
Half-life [h]	40–50	8	13–20
Route of elimination	hepatic	renal	hepatic
Unchanged [%]	10	1.4	<1

**Table 3 metabolites-10-00106-t003:** Antifungal drug dosages [mg/kg/day] for intravenous administration only during the initial therapy of invasive fungal infections in selected patient populations. IC = invasive candidiasis, IA = invasive aspergillosis, HSCT = patients under hematopoietic stem cell transplantation, CMTH = patients under chemotherapy, NN = non-neutropenic patients, NH = non-hematological patients.

Population	Neonatals (IC) [42]	Neonatals (IA) [43]	Children (IC) [48]	Children (IA) [43]	Adults (IC) [44]	HSCT (IA) [45]	CMTH (IA) [45]	NN (IA) [45]	NH (IA) [45]
AMBD	0.5–1.5	1.0–1.5	0.6–1.5	1.0–1.5	note ^1^	note ^1^	note ^1^	note ^1^	note ^2^
ABCD	3–5	note ^1^	note ^1^	note ^1^	note ^1^	note ^1^	note ^1^	note ^1^	note ^2^
ABLC	5	5	1–5	5	note ^1^	note ^1^	note ^1^	note ^1^	note ^2^
LAMB	1–5	3	1–5	3	3	3	3	3	note ^1^
5-FC	100	note ^2^	25–100	note ^2^	note ^2^	note ^2^	note ^2^	note ^2^	note ^2^
FLC	12	note ^2^	6–12	note ^2^	note ^1^	note ^2^	note ^2^	note ^2^	note ^2^
ITC	note ^2^	note ^3^	5–10	10	note ^1^	note ^2^	note ^1^	note ^2^	note ^2^
VOR	note ^3^	note ^3^	8–16	8–16	3–6	note ^1^	note ^1^	note ^1^	note ^1^
POS	note ^2^	note ^3^	note ^3^	note ^3^	note ^1^	2–12	8–12	2–12	note ^1^
ISV	note ^1^	note ^2^	note ^2^	note ^2^	note ^2^	note ^2^	note ^2^	note ^2^	note ^2^
ANF	1.5	note ^1^	note ^3^	note ^1^	100–200 ^4^	note ^1^	note ^2^	note ^1^	note ^2^
CSF	0.5–2	25–70 ^5^	50–70 ^5^	50–70 ^5^	50–70 ^4^	note ^1^	70 ^5^	note ^1^	note ^1^
MCF	4–15	note ^3^	2–4	note ^3^	100 ^4^	note ^1^	100 ^4^	note ^1^	note ^2^

^1^ drug has marginal or no recommendation for its use in this population. ^2^ drug not mentioned in the selected reference. ^3^ drug not yet approved for medical use (worldwide or European Union). ^4^ mg/kg. ^5^ mg/m^2^/kg.

**Table 4 metabolites-10-00106-t004:** The susceptibilities of antifungal drugs against selected *Aspergillus* species are expressed as minimal inhibitory concentrations inhibiting the growth of 90% of the microorganism (*MIC_90_*) [mg/L]. AA = antifungal agent, IE = insufficient evidence.

AA	*MIC_90_* [mg/L]									
	*Aspergillus*									
	*flavus*		*fumigatus*		*nidulans*		*niger*		*terreus*	
	S≤	R>	S≤	R>	S≤	R>	S≤	R>	S≤	R>
AMB	–	–	1	1	–	–	1	1	–	–
ANF	IE	IE	IE	IE	IE	IE	IE	IE	IE	IE
CSF	IE	IE	IE	IE	IE	IE	IE	IE	IE	IE
FLC	–	–	–	–	–	–	–	–	–	–
ISV	1	2	1	2	0.25	0.25	IE	IE	1	1
ITC	1	1	1	1	1	1	IE	IE	1	1
MCF	IE	IE	IE	IE	IE	IE	IE	IE	IE	IE
POS	IE	IE	0.125	0.25	IE	IE	IE	IE	0.125	0.25
VOR	IE	IE	1	1	1	1	IE	IE	IE	IE

**Table 5 metabolites-10-00106-t005:** The susceptibilities of antifungal drugs against selected *Candida* species are expressed as *MIC_90_* [mg/L]. AA = antifungal agent, IE = insufficient evidence.

AA	*MIC_90_* [mg/L]											
	*Candida*											
	*albicans*		*dubliniensis*		*glabrata*		*krusei*		*parapsilosis*		*tropicalis*	
	S≤	R>	S≤	R>	S≤	R>	S≤	R>	S≤	R>	S≤	R>
AMB	1	1	1	1	1	1	1	1	1	1	1	1
ANF	0.03	0.03	–	–	0.06	0.06	0.06	0.06	4	4	0.06	0.06
CSF ^1^	–	–	–	–	–	–	–	–	–	–	–	–
FLC	2	4	2	4	0.001	16	IE	IE	2	4	2	4
ISV	IE	IE	IE	IE	IE	IE	IE	IE	IE	IE	IE	IE
ITC	0.06	0.06	0.06	0.06	IE	IE	IE	IE	0.125	0.125	0.125	0.125
MCF	0.016	0.016	–	–	0.03	0.03	IE	IE	2	2	IE	IE
POS	0.06	0.06	0.06	0.06	IE	IE	IE	IE	0.06	0.06	0.06	0.06
VOR	0.06	0.25	0.06	0.25	IE	IE	IE	IE	0.125	0.25	0.125	0.25

^1^ EUCAST breakpoints have not yet been established (significant inter-laboratory variation).

**Table 6 metabolites-10-00106-t006:** Complete (green), ongoing (orange), terminated (blue), and not yet realized (gray) phases of clinical trials of new promising antifungals. More details about these drugs are in the following text. AA = antifungal agent.

AA	Identification	Last Update	Phase 0	Phase 1	Phase 2	Phase 3	Phase 4
VT-1129 [79]	–	–					
VT-1161 [80]	NCT03562156	5th Feb 2020					
VT-1598 [81]	–	–					
PC1244 [82]	–	–					
SUBA-ITC [80]	NCT03572049	25th Oct 2019					
CAMB [80]	NCT02971007	2nd Nov 2018					
Rezafungin [80]	NCT03667690	5th Feb 2020					
Ibrexafungerp [80]	NCT03363841	2nd Dec 2019					
Fosmanogepix [80]	NCT03604705	28th Feb 2020					
Olorofim [80]	NCT03583164	18th Jan 2020					
MGCD290 [80]	NCT01497223	4th April 2013					
T-2307 [83]	–	–					
VL-2397 [80]	NCT03327727	27th Feb 2019

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
