# Peer review of "Antifungal Drugs"

_metabolites, 2020, doi:10.3390/metabo10030106_

Round 1

Reviewer 1 Report

Abbreviations: Please write the abbreviations out in figure 1 - at least in the legend.
Although the abbreviations are very consistently used an occasional reminder (especially at the beginning of a chapter) might increase readability.

Figure 2 - bold instead of color would perhaps be better
line 127 introduce cytarabine

Figure 3: The graphic design has room for improvement.

Figure 4: red and green is not to be combined in a figure. Red–green color blindness affects up to 8% of males and 0.5% of females of Northern European descent.

Table 3 and 4: also a red - green contrast

The purpose of Figure 6 is unclear to me. It is referenced once, the legend is uninformative and the text does not really explain its significance. Perhaps a graph would do a better job - or more explanation.

line 12: a general emergence of antifungal resistance - is not dramatic if the reports are correctly interpreted. 

line 26 please rephrase "Inclusion of mucormycosis in India may result in 900 000 cases per year [4]."

line 222 "According to [59], ..." sounds strange

line 232 "develops" - perhaps rephrase
line 234 singular?

line 243 "The clinically relevant Fusarium spp." perhaps specify

line 249 explain "short-tailed"

line 254 - 256. perhaps rephrase these sentences
line 273 "Of note is the potential metabolic ..." please explain to the reader.
"A recent review [72] discussed ..." and came to which conclusions?

The "antifungal pipeline" chapter deserves a figure with the substances in question.
Especially this last chapter should be carefully checked for writing style and presentation of facts and their discussion. A conclusion on the pipeline and the current state of our armoury against fungi could conclude the text.

Author Response

Dear Editor, please, accept our apologies for the delayed response. On the other hand, our revisions have been much deeper. We re-mastered most of the figures, updated the tables a text and improved the manuscript clarity, readability and grammar. We do hope the reviewers will like it even better. Below, please, find our responses, that will be copy-pasted into the web portal. With best regards, Vladimir Havlicek, on behalf of all co-authors.

Review Report 1

  • Abbreviations: Please write the abbreviations out in figure 1 - at least in the legend.
  • Answer: Corrected. Figure 1 was re-mastered, also the most recent experimental drugs were added to the timeline. In addition, we changed the color coding for better visual clarity.  

  • Although the abbreviations are very consistently used, an occasional reminder (especially at the beginning of a chapter) might increase readability.
  • Answer: Corrected.

  • Figure 2 - bold instead of color would perhaps be better
  • Answer: Corrected

  • line 127 introduce cytarabine

Answer: Amended, we introduced cytarabine and added reference [30]: Co-administration of 5-FC with the cytarabine (an effective drug for the treatment of acute myeloid leukemia [30]) competitively inhibits its antifungal activity due to the same transport system by susceptible cells.

  • Figure 3: The graphic design has room for improvement.
  • Answer: We thank you for this comment, Figure 3 was re-mastered, blank space was removed. We reduced the size of this figure and deleted some parts. We divided the figure into the part A and B and added more information into the caption.

  • Figure 4: red and green is not to be combined in a figure. Red–green color blindness affects up to 8% of males and 0.5% of females of Northern European descent.
  • Answer: we absolutely agree and made the corresponding adjustments both to the Figures 4 and 5

  • Table 3 and 4: also a red - green contrast
  • Answer: From the original manuscript we subtracted Table 1. In the revision we created Tables 2 and 3. Therefore, the previous Tables 3 and 4 are now Table 4 (Aspergillus ) and Table 5 (Candida spp.). In both tables, we updated the colors in link to EUCAST 2019 breakpoints [reference 55].
  • The purpose of Figure 6 is unclear to me. It is referenced once; the legend is uninformative and the text does not really explain its significance. Perhaps a graph would do a better job - or more explanation.
  • Answer: We thank for this comment and admit it is a bit quirky. On the other hand, we want the reader to think about the links among resistance, factors and mechanisms. To improve the clarity, we added new text

  • line 12: a general emergence of antifungal resistance - is not dramatic if the reports are correctly interpreted.
  • Answer: Corrected - we changed the wording accordingly: The antifungal resistance may be developed across most pathogens and includes drug target overexpression, efflux pump activation, and amino acid substitution.

  • line 26 please rephrase "Inclusion of mucormycosis in India may result in 900 000 cases per year [4]."
  • Answer: We double-checked the literature and rephrased the sentence: The incidence of mucormycosis may exceed 900 000 cases per year after the inclusion of Indian data estimates [4].

  • line 222 "According to [59], ..." sounds strange
  • Answer: We changed the sentence structure: Furthermore, FCYB gene expression is repressed by the cytosine-cytosine-adenosine-adenosine-thymidine binding complex (CBC) and pH-response transcriptional factor PacC at pH 7, confirming intrinsic resistance [69].
  • Moreover, we made similar adjustments at the lines 27–30: Unfortunately, epidemiology of invasive fungal infections usually focuses on specific areas; therefore, the lack of available global data leads to the broad range of mortality rates, e.g. 30–95 % and 46–75 % in invasive aspergillosis and candidiasis, respectively [5].

  • line 232 "develops" - perhaps rephrase
  • Answer: We rephrased it accordingly: Furthermore, intrinsic resistance to triazoles is often caused by Candida biofilm formation [72].

  • line 234 singular?
  • Answer: “homologues” were changed to “homologue”

  • line 243 "The clinically relevant Fusarium spp." perhaps specify
  • Answer: Corrected, the new wording is “The clinically relevant Fusarium (especially F. solani complex, F. oxysporum, and F. fujikuroi complex [75]) are resistant to amphotericin B, triazoles, and echinocandins”.

  • line 249 explain "short-tailed"
  • Answer: In Figure 2, the short-tailed azoles are fluconazole and voriconazole, the mid-tailed isavuconazole, and the long-tailed itraconazole and posaconazole. We have rewritten the sentence and used the drug names: Mucormycetes are resistant to short-tailed triazoles, such as voriconazole and fluconazole.

  • line 254 - 256. perhaps rephrase these sentences

  • Answer: We agree and rephrased the text accordingly: “This higher-molecular weight cell wall content may play a role in echinocandin resistance”

  • line 273 "Of note is the potential metabolic ..." please explain to the reader.
  • "A recent review [72] discussed ..." and came to which conclusions?
  • Response: This paragraph was removed.

  • The "antifungal pipeline" chapter deserves a figure with the substances in question.
  • Answer: We agree and added Figure 7 with selected chemical structures of the substances mentioned in the chapter Antifungal pipeline.

  • Especially this last chapter should be carefully checked for writing style and presentation of facts and their discussion.
  • Answer: We rewrote this chapter (please, see the tracked changes).

  • A conclusion on the pipeline and the current state of our armoury against fungi could conclude the text.
  • Answer: we have rewritten the text and opted for a shorter but impactful wording.

Reviewer 2 Report

Houst et al. wrote a review about antifungal agents. There are more reviews in literature focusing on antifungal agents and this review can not add significantly more information to these reviews in this form. Nevertheless, good reviews are continuously needed. However, you must improve your manuscript significantly.

Minor points:

-The used abbreviations are very strange and unusual, please change them to generally accepted abbreviations (e.g.: AMB, CSF, MCF, ANF, VOR, FLC etc.)

-You should supplement the table 4. with C. dubliniensis breakpoints to azoles  (Eucast 2020).

-Make a table about pharmacological characteristics of echinocandins similar to azoles. 

Major points:

-It would be worth that EUCAST-related part would be supplemented with CLSI breakpoints or CLSI-related information  (similarity vs differences etc.).

-It is necessary to write about dosages focusing on given patient population in case of each drug (adult, paediatrics, pregnancy etc.).

-In the part of Antifungal pipeline, please add information about encochleated amphotericin B.

-Could you make a separate figure about the current clinical phase(s) of antifungal agents from the part of antifungal pipeline?

These improvements would significantly increase the quality of manuscript.

Others:

One-third of references is older than five years. 

English should be revised.

Author Response

Dear Editor, please, accept our apologies for the delayed response. On the other hand, our revisions have been much deeper. We re-mastered most of the figures, updated the tables a text and improved the manuscript clarity, readability, and grammar. We do hope the reviewers will like it even better. Below, please, find our responses, that will be copy-pasted into the web portal. With best regards, Vladimir Havlicek, on behalf of all co-authors.

Review Report 2

  • The used abbreviations are very strange and unusual, please change them to generally accepted abbreviations (e.g.: AMB, CSF, MCF, ANF, VOR, FLC etc.)
  • Response: we fully agree and we changed these abbreviations as following: amphotericin B = AMB, anidulafungin = ANF, caspofungin = CSF, 5-flucytosine = 5-FC, fluconazole = FLC, isavuconazole = ISV, itraconazole = ITC, micafungin = MCF, posaconazole = POS, voriconazole = VOR

  • You should supplement the table 4. with C. dubliniensis breakpoints to azoles  (Eucast 2020).
  • Response: From the original manuscript we deleted Table 1. In the revised version we created Tables 2 and 3. The previous Tables 3 and 4 now appear as Table 4 (Aspergillus ) and Table 5 (Candida spp.). We updated both tables according to EUCAST 2019-2020 breakpoints [reference 55] valid from 4th February 2020. Table 5 contains breakpoints for Candida dubliniensis.

  • Make a table about pharmacological characteristics of echinocandins similar to azoles.
  • Response: we fully agree and created new Table 2 which includes the analogous pharmacological properties similarly to Table 1. We added three new references to the Table 3 [39, 40, 41] and changed the text in respective paragraph.

  • It would be worth that EUCAST-related part would be supplemented with CLSI breakpoints or CLSI-related information (similarity vs differences etc.).
  • Response: we agree and modified the tex. CLSI distinguishes between SDD (susceptible, dose dependent) and NS (non-susceptible) category [53]. Despite the slight methodological differences, both CLSI and EUCAST yield comparable MIC data. Unluckily, CLSI has not determined the clinical breakpoints for any molds [54], so the data in Tables 4 and 5 summarize the susceptibilities of antifungal agents against selected Aspergillus and Candida species covering the period 2007–2019 from EUCAST only. These data have been modified from [55].

  • It is necessary to write about dosages focusing on given patient population in case of each drug (adult, paediatrics, pregnancy etc.).
  • Response: We again agree and added a new paragraph and Table 3 covering these niches.

  • In the part of Antifungal pipeline, please add information about encochleated amphotericin B.
  • Response: we thank for this important comment and added this information: “In addition to ergosterol sequestration, amphotericin B cochleate (CAMB) has been developed as an oral formulation of AMB. CAMB is made up of phosphatidylserine and phospholipid-calcium precipitates and demonstrates successful treatment in murine mouse model against C. albicans [85].”

  • Could you make a separate figure about the current clinical phase(s) of antifungal agents from the part of antifungal pipeline?
  • Response: This new information now appears as Table 6.

  • One-third of references is older than five years.
  • Response: we thank for this comment and deleted the obsolete references and added 2020 new ones.

  • English should be revised.
  • Answer: In the whole manuscript we revised the English (see tracked changes).

Round 2

Reviewer 2 Report

The quality of manuscript was significantly improved.